# Cross-Sectional Assessment on Carbapenem-Resistant Gram-Negative Bacteria Isolated from Patients in Moldova

**DOI:** 10.3390/microorganisms13020421

**Published:** 2025-02-14

**Authors:** Vadim Nirca, Frieder Fuchs, Tessa Burgwinkel, Rocío Arazo del Pino, Ecaterina Zaharcenco, Ralf Matthias Hagen, Sven Poppert, Hagen Frickmann, Paul G. Higgins

**Affiliations:** 1Molecular Biology Department, Imunotehnomed Ltd., MD-2001 Chisinau, Moldova; nirca.vadim@medexpert.md (V.N.); ecaterina.zaharcenco@medexpert.md (E.Z.); 2Department of Microbiology and Hospital Hygiene, Bundeswehr Central Hospital Koblenz, 56070 Koblenz, Germany; ralfmatthiashagen@bundeswehr.org; 3Landesuntersuchungsamt Rheinland-Pfalz, 56068 Koblenz, Germany; 4Institute for Medical Microbiology, Immunology and Hygiene, Faculty of Medicine, University Hospital Cologne, University of Cologne, 50931 Cologne, Germany; tessa.burgwinkel@uk-koeln.de (T.B.); rocio.arazo-del-pino@uk-koeln.de (R.A.d.P.); 5German Centre for Infection Research (DZIF), Partner Site Bonn-Cologne, 50935 Cologne, Germany; 6Bernhard Nocht Institute for Tropical Medicine Hamburg, 20359 Hamburg, Germany; sven@poppert.eu; 7Department of Microbiology and Hospital Hygiene, Bundeswehr Hospital Hamburg, 20359 Hamburg, Germany; frickmann@bnitm.de; 8Institute for Microbiology, Virology and Hygiene, University Medicine Rostock, 18057 Rostock, Germany

**Keywords:** epidemiology, carbapenem-resistance, Gram-negative bacteria, surveillance

## Abstract

Information on the molecular epidemiology and carbapenem resistance mechanisms in Gram-negative bacterial isolates in Moldova is scarce. To close this knowledge gap, carbapenem-resistant Gram-negative bacteria were collected over an 11-month period in a routine diagnostic laboratory in Moldova. Antimicrobial susceptibility was phenotypically and genotypically assessed. Phylogenetic relationships were investigated and multi-locus sequence types were provided. The assessment indicated several clusters of phylogenetically closely related carbapenem-resistant *Klebsiella pneumoniae* (sequence types ST101, ST395 and ST377), *Acinetobacter baumannii* (ST2, ST19 and ST78) and *Pseudomonas aeruginosa* (ST357 and ST654) isolates next to a number of less frequently observed species and sequence types. A phylogenetic relationship to characterized isolates from neighboring Ukraine could be confirmed. Identified carbapenemase genes comprised *bla*_OXA-23_, *bla*_OXA-72_ and *bla*_GES-11_ in *A. baumannii*, *bla*_KPC-3_, *bla*_NDM-1_ and *bla*_OXA-48_ in *K. pneumoniae*, as well as *bla*_VIM-2_ in *Pseudomonas aeruginosa*. In conclusion, the assessment suggested the spread of carbapenem-resistant Gram-negative bacteria in Moldova which were partly pre-described from neighboring Ukraine, as well as likely spill-over events, facilitating the regional spread of carbapenem-resistant clones. Several isolates with very high genomic similarity further support the hypothesis of likely regional transmission events driven by several evolutionary successful clonal lineages.

## 1. Introduction

Antibiotic resistance is an issue of relevance in the Republic of Moldova, not only regarding resistance in local *Mycobacterium tuberculosis* complex strains [1], but also in Gram-negative bacteria, including resistance to broad-spectrum beta-lactams [2,3]. In the recent “*Antimicrobial resistance surveillance in Europe 2023*” report by the ECDC (European Centre for Disease Prevention and Control) and WHO (World Health Organization) [3], locally reported carbapenem resistance for the last assessment year 2021 was 60.7% for *Klebsiella pneumoniae* complex, 73.1% for *Pseudomonas aeruginosa* and 95.2% for *Acinetobacter baumannii* complex. While geographical and hospital representativeness was considered as high for the ECDC/WHO surveillance, the authors admitted the limitation of low isolate representation [3]. Further, assessments on the molecular and epidemiological background of carbapenem resistance in Gram-negative bacterial isolates from Moldova are widely lacking.

Military conflicts are considered to facilitate the spread of multi-drug-resistant Gram-negative bacteria [4,5]. With respect to border movements of millions of people with or without medical needs along the borders of Ukraine and Moldova, it is likely that the military conflict in the neighboring Ukraine might influence the resistance profile in Moldova. Even in more distant Germany, a shift in the local carbapenemase distribution reported by the National Reference Centre could be attributed to patients from Ukraine [6].

High rates of carbapenem resistance were also reported earlier from Ukraine. In a global surveillance on meropenem resistance in *Escherichia coli* conducted from 2014 to 2021, Ukraine was in third place in the global ranking, after India and Pakistan, with a resistance rate of 5.4% [7]. While the problem was still considered as limited in a multinational surveillance program in 2011 [8], the situation changed with the onset of the regional conflict. During the period of the Euromaidan riots, a blood culture surveillance performed between 2013 and 2015 indicated a focus on non-fermenting Gram-negative bacteria. In this assessment, 33.1% *P. aeruginosa* isolates and 63.2% *A. baumannii* complex isolates were characterized as carbapenem-resistant [9]. From Ukrainian patients who were treated because of war wounds in Bundeswehr hospitals in Germany during this period of time, carbapenem-resistant *A. baumannii* isolates clustered with international clonal lineages IC1, IC2, IC6 and IC7, which corresponded to the Pasteur sequence types ST1, ST2, ST78 and ST25, respectively [10]. Identified molecular determinants of carbapenem-resistance among those isolates comprised the carbapenemase-encoding genes *bla*_OXA-23_, *bla*_OXA-72_ and *bla*_GES-12_ [10].

In the period from 2019 to 2021, the proportion of carbapenem-resistant non-fermentative Gram-negative bacteria isolated from Ukrainian inpatients during multi-centric surveillance had increased to 71.3% [11], while the carbapenem resistance rate was 19.1% for Enterobacterales [12]. In decreasing order of reported frequency, detection rates of carbapenem resistance-related beta-lactamases were 31.7% for *bla*_VIM_, 29.7% for *bla*_OXA_, 25.7% for *bla*_KPC_, 21.3% for *bla*_NDM-1_, and 5.7% for *bla*_IMP_, respectively [13]. Focusing on Ukrainian military hospitals within the time span 2014–2020, reported resistance rates against carbapenems in isolates of *A. baumannii*, *P. aeruginosa*, *K. pneumoniae* and *E. coli* were 67.9%, 55.6%, 32.8% and 42.9%, respectively. Co-expression of Ambler class A and D beta-lactamase genes was reported for *A. baumannii*, *P. aeruginosa* and *K. pneumoniae*, in the latter case associated with the Ambler class B beta-lactamase gene *bla*_NDM-1_ in a single instance [13]. Continuation of the surveillance during the conflict years 2022 and 2023 on isolates from war-injured patients indicated meropenem resistance in 72.2% *A. baumannii*, 90.6% *K. pneumoniae* and 47.8% *P. aeruginosa*. The *A. baumannii* sequence type ST-1077 emerged in addition to the previously prevalent sequence types ST78 an ST400, while ST-773 emerged among the *P. aeruginosa* isolates. Among the identified *K. pneumoniae*, an increase in the abundance of the beta-lactamase genes *bla*_NDM-1_, *bla*_OXA-48_ and *bla*_KPC2_ was recorded [14].

Apart from surveillance assessments in the Ukraine, recent surveillance on isolates from Ukrainian injured combatants in hospitals abroad led to the detection of carbapenemase genes like *bla*_NDM-5_, *bla*_KPC-3_, *bla*_OXA-48_, *bla*_NDM-1_ and other *bla*_NDM_ variants [15,16,17,18,19,20]. For non-fermentative Gram-negative bacteria from Ukrainian patients, in particular, *bla*_OXA-23_, *bla*_OXA-72_ and *bla*_NDM-5_ were reported for *A. baumannii* complex isolates [21] and *bla*_NDM-1_ for *P. aeruginosa* [22].

Considering the broad spectrum of carbapenem resistance mechanisms in neighboring Ukraine, this study intends to fill the knowledge gap on molecular resistance mechanisms in Moldova. For this purpose, carbapenem-resistant Gram-negative isolates were collected in a Moldovan diagnostic routine laboratory during an 11-month period and subjected to next-generation sequencing (NGS). By doing so, epidemiological information on both locally abundant resistance mechanisms and multi-locus sequence types was provided.

## 2. Materials and Methods

### 2.1. Study Design, Sample Collection and Available Epidemiological Information

The study was conducted as a mono-centric surveillance assessment at a Moldovan diagnostic routine laboratory located in Chisinau, receiving diagnostic samples from the surrounding area (Figure 1). Within the time period from January 2023 to November 2023, all diagnostically obtained carbapenem-resistant Gram-negative isolates (Enterobacterales and non-fermentative Gram-negative rod-shaped bacteria) were included in the assessment. The diagnostic procedures of the routine-laboratory comprised a full-spectrum cultural bacteriology in line with German industry standard (“Deutscher Industriestandard”) DIN EN ISO 15189, covering bacterial growth and isolation from all human tissues and body fluids. Differentiation and antimicrobial resistance testing were performed as described below. When phenotypic carbapenem resistance was detected, the Enterobacterales and non-fermentative Gram-negative rod-shaped bacteria were stored deep-frozen at −80 °C for subsequent NGS-based analysis. Only anonymously provided isolate-specific data comprising the isolation site of the human body, patient age, patient sex, in-patient or out-patient-status and geographic district of sample acquisition were obtained because ethical clearance for the presentation of more detailed patient-specific epidemiological data was denied as described below. Details are provided in the Appendix A Table A1.

### 2.2. Identification and Phenotypic Resistance Testing

All isolates were identified with MALDI-TOF and species identity was confirmed with next-generation sequencing (NGS) as detailed below. Antimicrobial susceptibility testing was performed as previously described comprising automated testing using the VITEK-II automate (bioMérieux, Nürtingen, Germany), disk diffusion with the Kirby–Bauer-method and microbroth dilution (Merlin, Bornheim-Hersel, Germany) in line with the recommendations provided in the interpretation standards of EUCAST (European Committee on Antimicrobial Susceptibility Testing) and as detailed by Sandford and colleagues [6]. Assessed antibiotic drugs comprised gentamicin and amikacin exemplarily for aminoglycosides, ciprofloxacin as a representative of the fluoroquinolones, cotrimoxazole, fosfomycin and the reserve antibiotics colistin and cefiderocol. Reduced susceptibility towards carbapenems was an inclusion criterion but is not specifically shown in the Section 3.

### 2.3. Next-Generation Sequencing

Carbapenem-resistant Enterobacterales and non-fermentative Gram-negative rod-shaped bacteria were grown overnight on blood agar plates and had their DNA extracted using the DNeasy UltraClean Microbial Kit (Qiagen, Hilden, Germany). The NEB Next^®^ Ultra™ II FS DNA Library Prep Kit for Illumina^®^ (New England Biolabs, Frankfurt, Germany) was used to prepare the libraries, which were subsequently sequenced for a 250 bp paired end in the Illumina MiSeq.

### 2.4. Bioinformatic Analysis

Genomes were assembled from the Fastq files using the SKESA assembler (Version 2.4.0) as part of the Ridom SeqSphere+ software (Ridom GmbH, Münster, Germany). Isolates with at least 96% targets were analyzed by core-genome MLST (cgMLST) using the validated schemes within SeqSphere (Version 10.0.05). Antimicrobial resistance genes were identified using NCBI AMRFinder Plus, which is also included in the SeqSphere software. Where applicable, seven-loci sequence types were extracted from the genomes.

### 2.5. Ethical Clearance

Ethical clearance for the presentation of detailed patient-specific data for a thorough epidemiological assessment was not granted to ensure the patients’ privacy. Accordingly, we abstained from presenting any patient-related information and focused on the bacterial isolates only, which is in line with Moldovan and German national law without requirement for ethical consultation. Strict compliance with the Declaration of Helsinki and all its amendments was ensured. The epidemiological data used are already public knowledge through the World Health Organization’s Global Antimicrobial Resistance and Use Surveillance System (GLASS) and the Central Asian and European Surveillance of Antimicrobial Resistance network (CAESAR), which the Moldovan laboratory MedExpert is part of in the global efforts to tackle antimicrobial resistance. The data are used with the patient’s written consent and in accordance with the Republic of Moldova’s national LAW no. 411 of 28 March 1995 on healthcare, LAW no. 263 of 27 October 2005 regarding the rights and responsibilities of the patient, LAW no. 133 of 8 July 2011 on the protection of personal data, LAW no. 264 of 27 October 2005 regarding the exercise of the medical profession, as well as the Moldavian Code of Ethics of the medical worker and pharmacist approved by the Government Decision no. 192 of 24 March 2017, meant to protect patients’ anonymity and privacy while satisfying the national needs for epidemiological surveillance and public health security.

## 3. Results

### 3.1. Sampling Sites, Epidemiological Information and Results of Phenotypic Resistance Testing

The isolates which were included in the assessment comprised *Acinetobacter baumannii* (*n* = 10), *Escherichia coli* (*n* = 1), *Klebsiella pneumoniae* (*n* = 51), *Providencia stuartii* (*n* = 1), *Pseudomonas monteilii* (*n* = 1), and *Pseudomonas aeruginosa* (*n* = 24). Details are provided in Table 1 and in the Appendix A Table A2. In descending numbers of isolates, medical facilities providing carbapenem-resistant isolates were located in the Moldovan districts Chisinau (*n* = 37), Orhei (*n* = 10), Bălți (*n* = 9), Cimislia (*n* = 7), Ialoveni (*n* = 4), Floresti (*n* = 3), Nisporeni (*n* = 3), Sîngerei (*n* = 3), Drochia (*n* = 2), Anenii Noi (*n* = 1), Basarabeasca (*n* = 1), Cahul (*n* = 1), Dubasari (*n* = 1), Edinet (*n* = 1), Fălești (*n* = 1), Hincesti (*n* = 1), Leova (*n* = 1), Soroca (*n* = 1), and Ungheni (*n* = 1). The patients were generally young with a mean age (±standard deviation SD) of 59.6 (±18.8) years of age, the female: male ratio was 1:2, the in-patient:out-patient ratio was 1:1.1 with 11 in-patients on the intensive care unit (ICU). While the Enterobacterales and *P. aeruginosa* isolates were predominantly collected from urine samples, most *A. baumannii* isolates were obtained from wound swabs. Apart from beta-lactams, high resistance rates were observed for aminoglycosides (30–100% gentamicin resistance), fluoroquinolones (100% ciprofloxacin resistance with the exemption of the single identified *Pseudomonas monteilii* isolate), and cotrimoxazole (80.4–100% resistance) (Table 1). Interestingly, considerable phenotypical resistance rates against cefiderocol were recorded for both *K. pneumoniae* (35.3%) and *P. aeruginosa* (45.8%). Colistin showed high susceptibility rates with the exception of three phenotypically resistant *K. pneumoniae* isolates.

### 3.2. Phylogenetic Trees and Detected Multi-Locus Sequence Types

The results of the *A. baumannii* sequence typing are summarized in Figure 2. Nine of the isolates clustered with established international clones (IC); three IC1 of which two were identical, two IC2 that with four alleles different can be considered as part of a transmission cluster, four IC6, of which two were identical, and a singleton, which was ST400.

There were four transmission clusters with the *P. aeruginosa* isolates (Figure 3). ST357 was the most common ST with 15 isolates and comprised two closely related transmission clusters. Interestingly these isolates did not possess a carbapenemase, and non-carbapenemase producers are not often associated with transmissions. The six ST654 isolates formed two transmission clusters and a singleton, and all carried the same carbapenemase, VIM-2. The remaining isolates were singletons.

The overwhelming majority of *K. pneumoniae* were ST395, which formed one large transmission cluster of 27 isolates and a smaller cluster of five isolates (Figure 4). Both clusters included identical isolates and multiple different carbapenem resistance mechanisms, suggesting a dynamic resistome and different sub-clones which are not resolved using cgMLST. In contrast, the isolates within the two remaining transmission clusters, ST101 and ST377, harbored the same carbapenemases, suggesting more clonal transmission.

The single *E. coli* isolate was ST648, *P. stuartii* was ST373, which is the same ST as isolate 23PS_RO04_R01 from Romania, and *P. monteilii* ST125.

### 3.3. Molecular Resistance Determinants

The detected antimicrobial resistance determinants are detailed in Table A2 in the manuscript’s Appendix A; the main findings are summarized in the following section. Focusing on carbapenemases as determinants of carbapenem resistance, *bla*_OXA-23_, *bla*_OXA-72_ and *bla*_GES-11_ genes were observed in *A. baumannii*. In *K. pneumoniae* isolates, *bla*_KPC-3_, *bla*_NDM-1_ and *bla*_OXA-48_ genes were identified. The carbapenemase gene *bla*_VIM-2_ was recorded in a minority of 6 out of 24 (25.0%) carbapenem-resistant *Pseudomonas aeruginosa*, while beta-lactamases specifically associated with carbapenem resistance were not recorded in the other *P. aeruginosa isolates*. The single *E. coli* isolate carried a *bla*_NDM-1_ gene, the *P. stuartii* isolate had a *bla*_OXA-48_ and a *bla*_NDM-1_ gene and the *P. monteilii* isolate harbored a *bla*_IMP-1_ gene. Other genes for broad- or narrow-spectrum beta-lactamases, including beta-lactamases mediating extended-spectrum beta-lactam resistance (ESBL) like *bla*_CTX-M-15_, were also abundant, as detailed in Table A2.

Apart from beta-lactamase genes, resistance determinants for other classes of antibiotic drugs were recorded as well. The highest diversity was observed for genes and mutations mediating aminoglycoside and ciprofloxacin resistance; details are provided in Table A1. For other antibiotic classes, the mediating resistance genes showed considerably less diversity. In detail, resistance against fosfomycin and chloramphenicol was mediated by the genes *catA1*, *catA3*, *catB3*, *catB7*, *catB8*, *cmlA1*, *cmlA5*, *floR*, and *floR2*, cotrimoxazole resistance by the genes *dfrA1*, *dfrA5*, *dfrA7*, *dfrB2*, *sul1*, *sul2*, and tetracycline resistance by the genes *tet(A)*, *tet(B)* and *tet(G)*. The detailed distribution is shown in Table A2.

## 4. Discussion

The study was conducted to assess the epidemiology of Gram-negative rod-shaped bacteria in Moldova with a laboratory-based single-center cross-sectional assessment. It led to a number of findings.

Most carbapenem-resistant Gram-negative bacteria comprised the Enterobacterales *K. pneumoniae* and the non-fermenters *A. baumannii* and *P. aeruginosa*. While most of the Enterobacterales and *P. aeruginosa* were isolated from urine, *A. baumannii* isolates were predominantly obtained from wound swabs. Virtually all isolates were from clinically relevant isolation sites, suggesting potential etiological relevance. For each species, several clusters of isolates were recorded with high genetic similarity, suggesting likely transmission events. However, neither this nor clinical courses could be monitored due to the ethical restrictions applying to the presented data.

Focusing on the recorded carbapenemase genes, *bla*_OXA-23_, *bla*_OXA-72_ and *bla*_GES-11_ were observed in *A. baumannii*, *bla*_KPC-3_, *bla*_NDM-1_ and *bla*_OXA-48_ in *K. pneumoniae*, as well as *bla*_VIM-2_ in a minority of carbapenem-resistant *Pseudomonas aeruginosa*. Fortunately, the *bla*_NDM-1_-carrying *P. aeruginosa* clone ST773, which was isolated from patients from neighboring Ukraine, could not be found [22,23]. Nevertheless, the study findings match well with reports on carbapenem-resistant Ukrainian isolates [11,12,13,14,15,16,17,18,19,20,21,24], making spill-over events from this neighboring country likely. Furthermore, IC2 *Acinetobacter* harboring *bla*_OXA-23_ is usually the predominant lineage/carbapenemase associated with conflicts; however, in the Eastern European region for reasons that are currently unknown, IC6/*bla*_OXA-72_ predominates [10,20,25]. However, the quantitative relevance of such regional spill-over is difficult to quantify, because the detected carbapanemases occur in other European regions as well. For example, and as summarized by van Duin and Doi [26], metallo-betalactamase-producing Enterobacterales had emerged in Europe before the onset of the present Ukrainian conflict with foci in Romania, Denmark, Spain and Hungary. The epi-center for the successful spread of *bla*_OXA-48_ genes in Europe has been localized to Turkey [26]. Nevertheless, the recently recorded shift in the local carbapenemase distribution in central European Germany [6], which had been epidemiologically attributed to patients from Ukraine, makes respective influences in neighboring Moldova at least highly likely.

Numerous different resistance determinants were identified, suggesting considerable antibiotic selection pressure on the isolates and explaining the observed high rates of phenotypic resistance against aminoglycosides, fluoroquinolones and co-trimaxozole. The high abundance of metallo-enzymes among the carbapenemases might at least partially explain the considerable proportion of observed phenotypic resistance against the reserve antibiotic cefiderocol [27]. In contrast, no resistance genes or mutations explaining the low rate of phenotypically observed colistin resistance among the *K. pneumoniae* isolates were identified. Interestingly, however, the cefiderocol and colistin resistance rates observed for Moldovan carbapenem-resistant *K. pneumaniae* isolates matches well with previously reported resistance rates from the Ukraine [19]. Of note, *bla*_NDM-1_-carrying *P. stuartii* isolates as observed among the Moldovan isolates had previously been published from the neighboring Ukraine [28] as well, although the isolate observed here showed a higher genetic similarity with a Romanian isolate as detailed above.

The observed international clonal lineages IC1, IC2 and IC6 of *A. baumannii* match well with historic reports on Ukrainian isolates [10]. The same applied to the here-observed *A. baumannii* sequence types ST2 and ST78 [10]. And at least one isolate of the *A. baumannii* sequence type ST400, which recently emerged in Ukraine [14], could be identified. Focusing on the *P. aeruginosa* isolates, it is noteworthy that the identified clones ST235, ST357 and ST654 are among pre-described international high-risk clones [29]. Our work shows a relevant proportion of them among the investigated carbapenem-resistant Moldovan *P. aeruginosa* isolates. Interestingly, this phenomenon was not observed for *K. pneumoniae* and *E. coli* [30,31].

Although more than 5% of *E. coli* isolates in neighboring Ukraine are known to be carbapenem-resistant, only a single resistant isolate was observed in the current study. The reason for this obvious discrepancy in spite of a close spatial relationship of the countries remains unknown. Interestingly, the recent “*Antimicrobial resistance surveillance in Europe 2023*” report by the ECDC (European Centre for Disease Prevention and Control) and the WHO (World Health Organization) [3] does not report carbapenem-resistant *E. coli* as a quantifiable issue of concern in Moldova.

Our study has a number of limitations. First of all, the ethical restrictions did not allow a thorough epidemiological study with the presentation of facility-, transmission- and disease-specific data. However, we identified regionally common resistance mechanisms in carbapenem-resistant Gram-negative isolates. With the exception of feces, the isolation sites of the isolates make etiological relevance associated with the isolation events highly likely, although all assessed bacterial species can be associated with either infection or harmless colonization in human individuals. Due to the lack of information on the clinical courses and infection status of the individual patients, we also abstained from the presentation of molecular virulence factors which could not have been correlated with clinical disease. Second, although isolates were geographically distributed due to the inclusion of different districts in central, northern and southern Moldova (see Figure 1 and Table A1), the single-centric design of the study and the limited number of isolates makes the results not necessarily representative for Moldova. Third, funding limitations of the investigator-initiated study allowed only a restricted surveillance period for this cross-sectional assessment. Furthermore, these funding constraints also prevented the addition of long-read sequencing, which might have provided reliable information on mobile genetic elements. Fourth, only limited clinical information was available for the laboratory-based analysis and ethical restrictions led to a lack of specific information on the medical facilities where the patients were treated. Therefore, it was not possible to discriminate focal hospital outbreak events from the regional spread of evolutionarily successful clones.

## 5. Conclusions

In spite of the above-mentioned limitations, this study provides insights into the regional distribution of carbapenem-resistant Gram-negative bacteria, especially near Chișinău, the capital of Moldova, and thus contributes to the scarce available regional surveillance data. Further, the molecular assessment indicates phylogenetic similarity of Moldovan and Ukrainian isolates, suggesting likely spill-over events between the neighboring countries. Evolutionary successful sequence types associated with regional spread in Moldova were considered as possible.

## Figures and Tables

**Figure 1 microorganisms-13-00421-f001:**
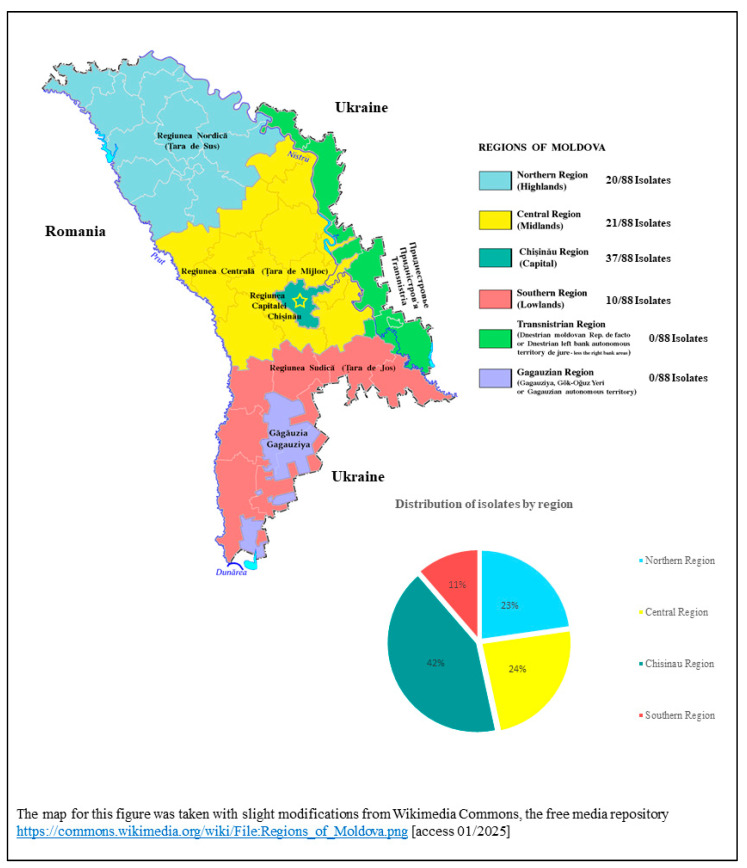
Map indicating the location of the diagnostic laboratory in Chisinau, Moldova (yellow star), and the regions from where samples were provided.

**Figure 2 microorganisms-13-00421-f002:**
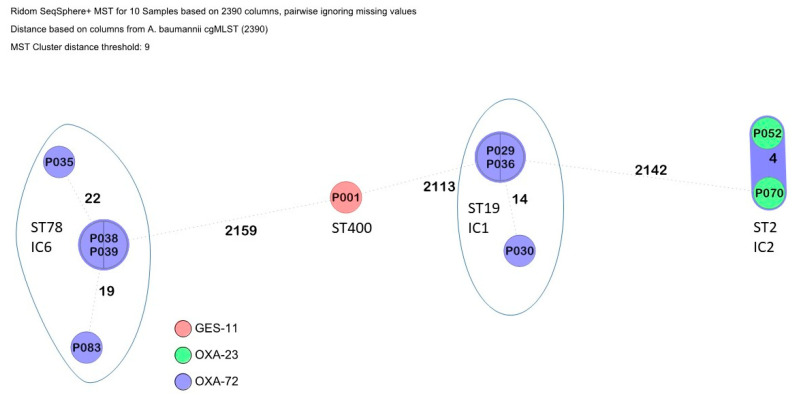
Clustering of the *A. baumannii* isolates. Carbapenemase carriage is indicated by color codes.

**Figure 3 microorganisms-13-00421-f003:**
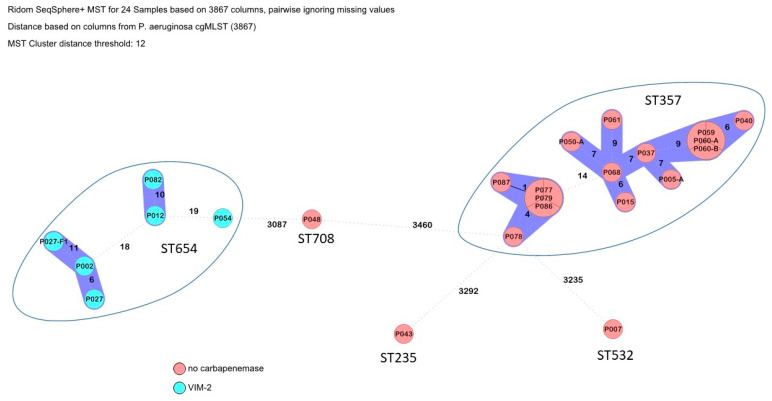
Clustering of *P. aeruginosa* isolates. Carbapenemase carriage is indicated by color codes.

**Figure 4 microorganisms-13-00421-f004:**
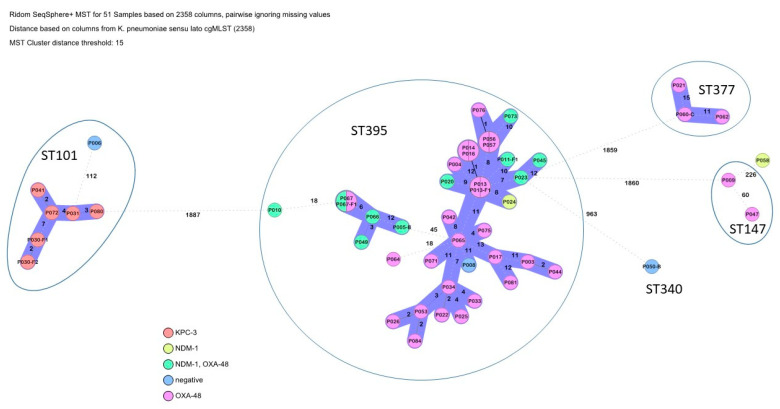
Clustering of the *K. pneumoniae* isolates. Carbapenemase carriage is indicated by color codes.

**Table 1 microorganisms-13-00421-t001:** Sampling sites and results of phenotypic resistance testing. Only resistance proportions in line with EUCAST interpretation standards are shown.

Species	Number (*n*)	Sampling Sites (%)	Percentage of Cefiderocol Resistance	Percentage of Gentamicin Resistance	Percentage of Amikacin Resistance	Percentage of Ciprofloxacin Resistance	Percentage of Trimethoprim/Sulfamethoxazole Resistance	Percentage of Colistin Resistance	Percentage of Fosfomycin Resistance
*Klebsiella pneumoniae* *	51	68.6% urine, 13.7% feces, 7.8% wound, 5.9% respiratory secretion, 2.0% blood, 2.0% bioptically taken body fluid	35.3%	68.0%	54.0%	100%	80.4%	5.7%	n.a.
*Pseudomonas aeruginosa*	24	64.0% urine; 12.0% respiratory secretion, 8.0% wound, 8.0% feces, 4.0% blood, 4.0% bioptically taken body fluid	45.8%	n.a.	72.0%	100%	n.a.	0%	n.a.
*Acinetobacter baumannii* *	10	60% wound, 30% respiratory secretion, 10% feces	n.a.	30.0%	20.0%	100%	100%	0%	n.a.
*Escherichia coli*	1	100% urine	100%	0%	0%	100%	100%	n.m.	0%
*Providencia stuartii*	1	100% urine	0%	100%	100%	100%	100%	n.a.	n.a.
*Pseudomonas monteilii* °	1	100% feces	0%	n.a.	n.m.	0%	n.a.	0%	n.a.

*n*. = number. n.a. = not applicable due to lack of EUCAST breakpoints. n.m. = not measured. * MALDI-TOF-MS allowed discrimination on the species-complex level only. ° The isolate was initially misclassified by MALDI-TOF-MS as *Pseudomonas putida* and identified from the genome sequence using JSpeciesWS (version 4.2.1).

## Data Availability

All relevant information is provided in the manuscript and its appendices. Raw data can be made available at reasonable request. Raw reads were submitted to the ENA under the project number PRJEB81895.

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
