# Peer review of "Cross-Sectional Assessment on Carbapenem-Resistant Gram-Negative Bacteria Isolated from Patients in Moldova"

_microorganisms, 2025, doi:10.3390/microorganisms13020421_

Round 1
Reviewer 1 Report
Comments and Suggestions for Authors
The authors evaluated the occurrence of carbapenemase-producing Gram-negative bacteria in a healthcare facility (monocentric surveillance) in Moldova and investigated the underlying mechanisms of resistance, as well as the population structure of the isolates using whole genome sequencing (WGS), a powerful method for studying antimicrobial resistance (AMR).
While the study is monocentric, its findings are important and relevant to the public health community at both local and global levels. I have a few suggestions to further enhance the manuscript:
1) Provide detailed information about the healthcare facility, including a map, to offer context for the study.
2) Include information about the patients who carried these pathogens, such as demographic details, clinical data, outpatient or in-patient, and risk factors, to strengthen the clinical relevance of the findings.
3) Present data on the prevalence of carbapenemase-producing and carbapenem-resistant Gram-negative bacteria in the studied population.
4) Provide additional information on the plasmid types, as well as human pathogenicity and virulence factors.
5) Consider adding a phylogenetic tree for the isolates, particularly for Klebsiella, Pseudomonas aeruginosa, and Acinetobacter baumannii. While the clustering figures are informative, a phylogenetic tree would offer a clearer representation of genetic relationships. Also, clarify whether the isolates are nosocomial or community-acquired, and if there are any observed outbreaks.
6) It is surprising that only one E. coli isolate was identified, as this species is typically predominant in hospital-acquired infections. Could the authors provide an explanation for this?
7) Please include any discrepancies between the MALDI-TOF results and WGS findings in Table A1.
8) Provide a summary of the common resistance patterns observed in carbapenem-resistant Gram-negative bacteria.
9) The discussion section should be expanded to include a more thorough comparison with data from Europe.
10) Improve the conclusion section and make it more realistic. Given that this is a monocentric study, the conclusions should reflect the limitations of the study design, and avoid overgeneralizing the findings to the national level.
Author Response
Reviewer 1, first comment:
The authors evaluated the occurrence of carbapenemase-producing Gram-negative bacteria in a healthcare facility (monocentric surveillance) in Moldova and investigated the underlying mechanisms of resistance, as well as the population structure of the isolates using whole genome sequencing (WGS), a powerful method for studying antimicrobial resistance (AMR).
While the study is monocentric, its findings are important and relevant to the public health community at both local and global levels. I have a few suggestions to further enhance the manuscript:
Provide detailed information about the healthcare facility, including a map, to offer context for the study.
Authors:
As requested, we have stated that the assessment was conducted at a Moldovan diagnostic routine laboratory located in Chisinau receiving diagnostic samples from the surrounding area (Methods chapter, sub-heading “Study design, sample collection and available epidemiological information”,1st sentence). A map of the region from where the samples were shipped is now provided as a new figure 1, further details on the respective districts are provided in a new appendix Table A1. More details (e.g. on clinical facilities) cannot be provided due to ethical restrictions as now explained in more detail in the limitations chapter of the discussion (discussion, last paragraph, first limitation) and in the revised information for ethical clearance in the manuscript.
Reviewer 1, second comment:
Include information about the patients who carried these pathogens, such as demographic details, clinical data, outpatient or in-patient, and risk factors, to strengthen the clinical relevance of the findings.
Authors:
In spite of the limitations due to the ethical restrictions of this study, we have now tried to address the point as differentiated as possible in the results (sub-heading “Sampling sites, epidemiological information and results of phenotypic resistance testing”, new second and third sentence) as well as in the limitations paragraph of the discussion (discussion, last paragraph, first limitation). In detail, we have elaborated that the ethical restrictions did not allow a thorough epidemiological study with the presentation of facility-, transmission- and disease-specific data. However, we identified regionally common resistance mechanisms in carbapenem-resistance Gram-negative isolates. Further, the isolation sites of the isolates make etiological relevance associated with the isolation events highly likely, although all assessed bacterial species can be associated with either infection or harmless colonization in human individuals. However, in line the ethical requirements, we were able to state that In descending numbers of isolates, medical facilities providing carbapenem-resistant isolates were located in the Moldovan districts Chisinau (n=37), Orhei (n=10), BălÈ›i (n=9), Cimislia (n=7), Ialoveni (n=4), Floresti (n=3), Nisporeni (n=3), Sîngerei (n=3), Drochia (n=2), Anenii Noi (n=1), Basarabeasca (n=1), Cahul (n=1), Dubasari (n=1), Edinet (n=1), FăleÈ™ti (n=1), Hincesti (n=1), Leova (n=1), Soroca (n=1), Ungheni (n=1). The patients were generally young with a mean age (± standard deviation SD) of 59.6 (± 18.8) years of age, the female : male ratio was 1 : 2, the in-patient : out-patient ratio was 1 : 1.1 with 11 in-patients on the intensive care unit (ICU). Details were also provided in the new appendix Table A1 and we have elaborated in the ethics chapter of the methods on the reasons of the presentation of restricted details.
Reviewer 1, third comment:
Present data on the prevalence of carbapenemase-producing and carbapenem-resistant Gram-negative bacteria in the studied population.
Authors:
Due to the fact that affiliation- and patient-specific information could not be used for ethical reasons, there is no denominator for the calculation of prevalences. Available information on carbapenem-resistance in Moldovan patients in general was already provided in the introduction (first paragraph, second sentence). Due to the ethical restriction, the provided study is on regional carbapenem-resistant isolates, not on their distribution within the local population.
Reviewer 1, fourth comment:
Provide additional information on the plasmid types, as well as human pathogenicity and virulence factors.
Authors:
We had considered this but abstained from doing so for the following reasons, which we have now detailed in the limitations paragraph (last paragraph of discussion). Regarding the virulence factor analysis, we have now elaborated that due to the lacking information on clinical courses and infection status of the individual patients, we abstained from the presentation of molecular virulence factors which could not have been correlated with clinical disease (discussion, last paragraph, 4th sentence). You will be aware of the fact that there are multiple molecular virulence determinants and without a comparison with clinical information, such an approach would just lead to loads of difficult to interpret information. Regarding the mentioned plasmids, we have stated that funding constraints also prevented the addition of long-read sequences, which might have provided reliable information on mobile genetic elements (discussion, last paragraph, last-but-one sentence). Without long-read sequences, we’d like to abstain from providing information on mobile genetic elements for reasons of quality control. We respectfully ask the editor to accommodate our respective decisions.
Reviewer 1, fifth comment:
Consider adding a phylogenetic tree for the isolates, particularly for Klebsiella, Pseudomonas aeruginosa, and Acinetobacter baumannii. While the clustering figures are informative, a phylogenetic tree would offer a clearer representation of genetic relationships. Also, clarify whether the isolates are nosocomial or community-acquired, and if there are any observed outbreaks.
Authors:
Again, we respectfully ask the editor to accommodate our decision of maintaining the present cluster-based visualization, as we do not feel that another visualization strategy will relevantly add to the contents of the manuscript. In particular, a phylogenetic tree will not clarify the difference between nosocomial- and community-acquired isolates considering the non-availability of clinical data. Further, the provided multi-locus sequence typing already shows the relationship between isolates clearly and easily. We have also more clearly indicated the limitation in the data interpretation based on ethical restrictions as a last sentence of the discussion section (line 346-348).
Reviewer 1, sixth comment:
It is surprising that only one E. coli isolate was identified, as this species is typically predominant in hospital-acquired infections. Could the authors provide an explanation for this?
Authors:
We have now addressed this point in the discussion (6th paragraph). In detail, we have now stated that although more than 5% of E. coli isolates in neighboring Ukraine are known to be carbapenem-resistant, only a single resistant isolate was observed in the assessment presented here. The reason for this obvious discrepancy in spite of close spatial relationship of the countries remains unknown. Interestingly, the recent “Antimicrobial resistance surveillance in Europe 2023” report by ECDC (European Centre for Disease Prevention and Control) and WHO (World Health Organization) does not report carbapenem-resistant E. coli as a quantifiable issue of concern in Moldova as well.
Reviewer 1, seventh comment:
Please include any discrepancies between the MALDI-TOF results and WGS findings in Table A1.
Authors:
As requested, we have now added respective information, but in the footnote of table 1 rather than in Table A1, which details on the detected resistance-associated genes. In the footnote of table 1, we have now clarified that MALDI-TOF-MS allowed discrimination of A. baumannii and K. pneumoniae on the species-complex level only and that the Pseudomonas monteilii isolate was initially misclassified by MALDI-TOF-MS as Pseudomonas putida. Altogether, we did not consider these discrepancies as severe.
Reviewer 1, eighth comment:
Provide a summary of the common resistance patterns observed in carbapenem-resistant Gram-negative bacteria.
Authors:
We agree. Although the phenotypically detected resistance patterns had already been presented in table 1, we have now added the most relevant findings in brackets in the main text (Results chapter, sub-heading “Sampling sites and results of phenotypic resistance testing”, 4th and 5th sentence).
Reviewer 1, nineth comment:
The discussion section should be expanded to include a more thorough comparison with data from Europe.
Authors:
We have now addressed this topic in the discussion (3rd paragraph, 4th to 7th sentence). We agree that the focus has been too strict on comparisons with neighboring Ukraine. Accordingly, we have now provided some historic background on the spread of the detected carbapenemases in Europe. In detail, we have now stated that the quantitative relevance of regional spill-over from neighboring Ukraine is difficult to quantify, because the detected carbapanemases occur in other European regions as well. For example, and as summarized by van Duin and Doi, metallo-betalactamase-producing Enterobacterales had emerged in Europe before the onset of the present Ukrainian conflict with foci in Romania, Denmark, Spain and Hungary. The epi-center for the successful spread of blaOXA-48 genes in Europe has been localized in Turkey. Nevertheless, the recently recorded shift in the local carbapenemase distribution in central European Germany, which had been epidemiologically attributed to patients from Ukraine, makes respective influences in neighboring Moldova at least highly likely.
Reviewer 1, tenth comment:
Improve the conclusion section and make it more realistic. Given that this is a monocentric study, the conclusions should reflect the limitations of the study design, and avoid overgeneralizing the findings to the national level.
Authors:
We agree and have weakened the conclusions both in the abstract and in the main text accordingly.
Reviewer 2 Report
Comments and Suggestions for Authors
Title: Cross-sectional assessment on carbapenem-resistant Gram-negative bacteria isolated from patients in Moldova
The manuscript presented new data of significant importance.
Comments
- The English language should be improved
- Line 27-28: rewrite this sentence
- In Material:
- * Line 108-114: Rewrite this part. provide the type of collected samples, isolation, identification procedures and species of bacterial isolate.
- * Line 131: mention the species of bacteria you did next generation sequencing
- Section 3.3. improve the title and rewrite this section to be clearly understood.
- Please note the other comments in the attached PDF.

The English language should be improved
Author Response
Reviewer 2, first comment:
Title: Cross-sectional assessment on carbapenem-resistant Gram-negative bacteria isolated from patients in Moldova
The manuscript presented new data of significant importance.
Comments
The English language should be improved
Authors:
Prior to re-submission, thorough language proof-reading by the senior author Paul Higgins, who is a native English speaker, was performed to ensure appropriate language quality.
Reviewer 2, second comment:
Line 27-28: rewrite this sentence
Authors:
As requested, the sentence was split to make it better readable.
Reviewer 2, third comment:
In Material:
Line 108-114: Rewrite this part. provide the type of collected samples, isolation, identification procedures and species of bacterial isolate.
Authors:
We agree and have broadened the provided information accordingly. In detail, we have now stated that within the time period from January 2023 to November 2023, all diagnostically obtained carbapenem-resistant Gram-negative isolates (Enterobacterales and non-fermentative Gram-negative rod-shaped bacteria) were included in the assessment. The diagnostic procedures of the routine-laboratory comprised a full-spectrum cultural bacteriology in line with German industry standard (“Deutscher Industriestandard”) DIN EN ISO 15189, covering bacterial growth and isolation from all human tissues and body fluids. Differentiation and antimicrobial resistance testing were performed as described under the sub-heading below.
Reviewer 2, fourth comment:
Line 131: mention the species of bacteria you did next generation sequencing
Authors:
We agree and have clarified that Enterobacterales and non-fermentative Gram-negative rod-shaped bacteria were meant.
Reviewer 2, fifth comment:
Section 3.3. improve the title and rewrite this section to be clearly understood.
Authors:
We have assumed that you meant an adaption in line with the added PDF-file which you have attached and adapted the paragraph accordingly.
Reviewer 2, sixth comment:
Please note the other comments in the attached PDF.
Authors:
Your other deletions and wording adaptations were also included in the manuscript file in line with your suggestions.
Reviewer 3 Report
Comments and Suggestions for Authors
In this manuscript, authors present in detail epidemiology of carbapenem - resistant gram -negative pathogens in Moldova. The isolates were analysed by up to date molecular techniques such as next generation sequencing and resistance to antimicrobial agents was assessed. The manuscript is well written, conclusions are well supported by experimental data and findings are presented in association to existing data.
Line 158 A. baumannii delete r.
Author Response
Reviewer 3
In this manuscript, authors present in detail epidemiology of carbapenem - resistant gram -negative pathogens in Moldova. The isolates were analysed by up to date molecular techniques such as next generation sequencing and resistance to antimicrobial agents was assessed. The manuscript is well written, conclusions are well supported by experimental data and findings are presented in association to existing data.
Line 158 A. baumannii delete r.
Authors: done
We are grateful to the reviewers for their suggestions and hope that the manuscript is of interest for the readers of MDPI Microorganisms.
Sincerely,
Frieder Fuchs
Round 2
Reviewer 2 Report
Comments and Suggestions for Authors
The authors responded to all comments, no additional comments are required